# Informing State-Wide Coalition Efforts to Implement and Integrate Nutrition Best Practices in Early Care and Education: Focus Group Insights from Child Care Providers

**DOI:** 10.3390/ijerph191610025

**Published:** 2022-08-14

**Authors:** Brenda D. Koester, Stephanie Sloane, Sarah Chusid, Janna Simon

**Affiliations:** 1Family Resiliency Center, Department of Human Development and Family Studies, University of Illinois Urbana-Champaign, Urbana, IL 61801, USA; 2Department of Communication, University of Illinois Urbana-Champaign, Urbana, IL 61801, USA; 3Illinois Public Health Institute, Chicago, IL 60607, USA

**Keywords:** early care and education, nutrition best practices, Child and Adult Care Food Program, breastfeeding, screen time, implementation

## Abstract

A healthy diet in early childhood is an important contributor to ensuring lifelong health and in reducing risk for obesity. The child care environment is critical to supporting nutrition as a majority of young children less than 5 years of age are enrolled in out-of-home care. In order to better understand barriers to implementing and integrating nutrition best practices, we conduced focus groups with child care providers (*n* = 25) in Illinois. Providers from low-income communities, rural communities, and communities of color were prioritized. Focus group participants reported several challenges including the high cost of nutritious food, picky eating, and their perception that parents did not set good examples at home. Many providers identified the Child and Adult Care Food Program (CACFP) as a critical resource in helping them implement best practices. Providers discussed needing and wanting more training, more money for food, and more parental support. These results indicate support for additional resources and sustained training and technical assistance to address perceived challenges. The evidence of the importance of CACFP in helping providers engage in nutrition best practices indicates support for expansion and strengthening of the program.

## 1. Introduction

Poor diet is linked to diseases that are among the leading causes of death in the United States and that create billions of dollars in healthcare costs [1]. One of these diseases is obesity, which remains at a stubbornly high level for children aged 2 to 5 [2,3,4]. Children from minority and low-income families are at even greater risk for obesity [5,6]. One of the strongest predictors of adult obesity is having been obese in childhood [7]. Rates of adult obesity are predicted to continue to rise with one estimate indicating that by 2030 one in two adults will be obese [8]. This challenge has become even more critical given the impact of the COVID-19 pandemic on obesity rates [9,10,11,12].

The causes of childhood overweight and obesity are multifaceted and include influences that range from genetic to environmental [13]. Recommendations by the Institute of Medicine of the National Academies to prevent obesity in young children include ensuring access to a healthy diet and nutrition and supporting breastfeeding [14]. Consuming a poor diet is associated with greater risk of childhood obesity [15]. Taking a systems approach has been suggested as an effective way to address obesity [16] and, specifically, improving nutrition and access to a healthy diet in early childhood [17].

The complex systems approach is emerging as a way to think about addressing public health challenges, such as childhood obesity, that are driven by multiple spheres of influence [18,19]. This approach encourages learning directly from communities and individuals to address these challenges, and becomes even more important when working with historically disinvested communities. Once effective evidence-based interventions and policy changes have been identified, the existing challenges of implementation that can vary across context and setting still remain [20].

The Illinois State Physical Activity and Nutrition (ISPAN) program is an initiative funded by the Centers for Disease Control and Prevention (CDC) to implement physical activity and nutrition interventions designed to tackle the root causes of obesity and make it easier for Illinoisans to live the healthiest lives possible [21]. ISPAN focuses on low-income and rural communities, and communities of color that have been hardest hit by chronic diseases such as type 2 diabetes and heart disease. To reach their goal, the Illinois Public Health Institute (IPHI) convenes collaborators including the Illinois Department of Public Health (IDPH), local health departments in three regions of the state, and a coalition of organizations from multiple sectors to implement the ISPAN strategies.

ISPAN works through four strategies, only two of which are pertinent to this study and are discussed here. The first focuses on early care and education (ECE) by integrating nutrition standards and best practices in early childhood education systems. ECE settings are critical in efforts to prevent obesity [22,23] as a majority of children under age 5 are in some form of out-of-home care [24]. While there is still a need to further develop the evidence base, high-impact interventions and policy changes that are effective in improving and promoting healthy nutrition [22,25] have been identified. All states that received SPAN program funding from the CDC were encouraged to use Nemours’ Spectrum of Opportunities to implement these high-impact interventions [26]. Interventions and efforts that target multiple areas, or spheres of influence have been found to be more effective in improving the nutritional environment in ECE settings [27,28].

The second ISPAN strategy focuses on breastfeeding support by increasing the number of and connections between organizations that support new parents’ desires to breastfeed their babies. Addressing breastfeeding in early care settings is vital as child care providers can serve as an important source of support and information for parents who choose to breastfeed [29]. There is evidence that some parents do not feel supported [30,31] and that providers may not be comfortable offering advice [32].

In order to inform coalition and ISPAN efforts, IPHI and researchers from the University of Illinois conducted a series of focus groups with child care providers. Focus groups were utilized to elicit the widest possible range of experiences and opinions from child care providers across Illinois from five areas of the state representing rural, urban, and suburban regions. The specific goal of the focus groups was to better understand the needs, implementation practices, and barriers encountered by Illinois early child care providers around nutrition, physical activity, and breastfeeding.

## 2. Methods

Focus groups are ideal for this type of research because they are dynamic and interactive and yield data that is richer than what can be gathered by individual interviews because they enable participants to question and explain their answers to each other [33,34,35,36]. The focus group questions and protocol were developed by the research team and approved by the University of Illinois’s Institutional Review Board. Participants were recruited through Illinois Network of Child Care Resource and Referral Agencies (INCCRRA), local Child Care Resources and Referral (CCRR) staff, University of Illinois Extension, local health departments, Service Employees International Union (SEIU) Healthcare IL, and other state ECE partners. To qualify for the study, participants had to (1) currently work as a family, group home, or child care center owner, director, or teacher, (2) have worked for a minimum of one year in their current position, and (3) be 18 years or older. Eligibility was confirmed by a CCRR staff member.

Five focus groups were conducted from August to October 2020, with a total of 26 participants. Focus groups were held with providers based in Peoria (5), Cook County (4), West Chicago (6), Lake County (6; Spanish-speaking only), and Champaign (5). In four of the five focus groups, providers had previous experience providing care for infants but did not currently have infants in their care. One focus group was recruited explicitly to include providers currently caring for infants. Another focus group included Spanish-speaking-only providers. Focus groups were originally scheduled to be conducted in person but due to the stay-at-home order in response to the COVID-19 global pandemic [37], all focus groups were moved to an online format using Zoom teleconferencing software.

Participants were emailed a consent form and had the opportunity to ask questions with a member of the research team in a private Zoom breakout room at the beginning of the focus group session. Informed consent was obtained from all subjects involved in the study. Each focus group was led by a trained facilitator and had one note-taker from the research team present. Focus groups lasted 90 min on average and followed a similar format, with questions focusing on four areas: (1) implementing nutrition best practices, (2) supporting breastfeeding, (3) implementing physical activity best practices (not addressed in this paper), and (4) need for support and professional development. At the end of each focus group, the facilitator read a summary of impressions and conclusions to participants and asked them to confirm that they were accurate. Participants were then emailed a $40 USD Walmart or Amazon gift card.

Focus groups were audio-recorded using the recording feature on Zoom and professionally transcribed to increase accuracy [38]. Transcripts were analyzed by a team of trained qualitative researchers using a semantic approach to thematic analysis [39]. The semantic approach provides a framework to identify, interpret, and summarize the explicit meanings of the data rather than underlying constructs [39]. Dedoose web application [40] was used to manage, excerpt, and code the data. The research team read all transcripts individually and then worked together to develop a preliminary coding structure. The primary coder used the agreed-upon structure to code the remaining transcripts, while still allowing new codes to emerge. The secondary analyst used the complete coding structure to code an overlapping 25% of the transcripts. The analysis team discussed any disagreements until consensus was reached [41]. See Appendix A for codes, descriptors, and quote examples.

## 3. Results

### 3.1. Demographics

Focus group participants (*n* = 25) were all female and were an average of 44 years old (range 28 to 62). Thirty-six percent of participants self-identified as Hispanic or Latino individuals (*n* = 8), 50% of participants identified as Black individuals (*n* = 11), 36% of participants identified as White individuals (*n* = 8), 4.5% of participants identified as individuals of multiple races (*n* = 1), and 9% of participants selected “other” (*n* = 2). Participants were experienced and had worked in child care for an average of 15 years (range 1 to 36 years). The majority of participants worked in a family child care home (73%; *n* = 16), 23% worked in a child care center (*n* = 5), and 5% worked in a Head Start facility (*n* = 1). The majority of participants identified as family daycare providers (*n* = 11), 35% identified as classroom teachers (*n* = 8), and 17% as administrators (*n* = 4).

### 3.2. Implementing and Supporting Nutrition Best Practices

Focus group facilitators began by asking participants what they did to support the nutritional health of children in their care and how they became interested in doing so. Most providers across the focus groups said that licensing standards were not the primary influence in their decision-making around healthy nutrition practices. One provider said that licensing standards were a driving force initially but now it was mostly experience. Many providers talked about personal motivation as a major influence on their desire to support children’s nutritional health. Participants talked about their love and passion for children and wanting to keep them healthy. Providers discussed personal experiences with illness, such as having had cancer, their own struggle with weight, or being older as specific motivating factors. Some providers also mentioned being motivated by the personal results they saw after following the Child and Adult Care Food Program (CACFP) guidelines for nutrition. The CACFP program is a federally funded program that provides supplemental funding for providing healthy snacks and meals to eligible children in center- and home-based care. CACFP requires participants to serve meals/snacks that align with CACFP meal patterns, which are based on the Dietary Guidelines for Americans [42].

Participants were then asked about what helped them implement and support nutrition best practices. CACFP was mentioned by most providers as having played a significant role. Providers said that CACFP helped them with correct portioning of food, knowing what to buy at the grocery store, and finding menu ideas.

“*The nutrition program (CACFP), and the newsletter and the training they have there. That has really helped me to be able to implement, you know, have my thought, my thinking about nutrition.*”(Participant 1.2)

“*The food program (CACFP) because they have a lot of healthy meals just even on the calendar. You can look on there, there is a lot of resources on healthy meals.*”(Participant 1.3)

A few mentioned a “spillover” effect of CACFP to the child’s home with healthier nutrition practices being adopted by parents of children in their care. For some providers, parents played a role in their decision making around health practices, such as providing input on snack selection and nutrition activities. A few center-based providers mentioned having parent groups or soliciting parental input through regular parent surveys.

“*I totally agree with my colleagues. And I think that when we started nutrition practices in the meal program, education for us was reinforced a lot… knowing how to combine foods, requiring a healthier eating standard where everything has to be balanced. I totally agree with (other participant). Modifications soon began to be made especially for the children in our care, but eventually it began to be reinforced in the personal nutrition of our families. What helped me a little more with the parents of my program was sharing recipes. Sharing photos, sending them photos that their children are really trying these foods because many parents tell me, ‘I can’t believe my children eat this vegetable because at home we don’t consume it.’ And it is a way of showing the parents that, if they (their children) eat it, that they gladly eat them, that they are choosing to do so, that they are talking about the food with their other peers, and it is also a way that they (the parents) also begin to promote this diet at home with their child, reinforcing the effort that we are making.*”(Participant 4.2)

Providers found creative ways to implement and incorporate best practices into their child care. Tips and tricks included both creativity with offering foods (e.g., using fruit as dessert) and healthy cooking methods (e.g., using air fryers, roasting food).

“*Well I could say at first I had a challenge with the whole grain pasta. So I mixed it together so they wouldn’t be able to tell the difference. You know so I just added the regular pasta that they were used to eating and now it’s ok.*”(Participant 1.3)

Participants discussed ways that they interacted with children, such as involving them in food preparation and modeling healthy eating behaviors. These strategies were mentioned as ways to help kids become less picky about trying new foods.

“*We sit down as teachers and model that same behavior (“take a polite bite”), you know we’re eating the vegetables. We’re eating whatever the kids don’t like. Let them see what we do and try to ask them to take a ‘polite bite’ of the things they don’t like.*”(Participant 2.2)

### 3.3. Challenges to Nutrition Best Practices

Facilitators asked participants about challenges that they faced in implementing and supporting nutrition best practices. The cost of purchasing healthy food was a challenge that providers repeatedly mentioned as a barrier to adhering to nutritional best practices.

“*I think it just goes back again to the prices in the expense, because in the summer you can get such a big variety or fruits and vegetables, there’s no issues, but once it starts getting cold kind of start running into limits of what you can get, that’s not going to cost an arm and a leg. So in the summer time is fantastic. I can buy anything for cheap and the kids get beautiful selections as it gets to winter I kinda have to, you know, budget a little bit more and pay attention to it a little bit more. And I’m not gonna spend a ton of money and things that I’m not sure they like, so I’ll be a little bit less willing to try things with the kids in the winter when it’s more expensive.*”(Participant 1.5)

Another common barrier that was mentioned repeatedly by participants was children not wanting to try new foods and other picky eating behaviors. 

“*You know the kids, if they’re not used to eating something. Sometimes they look at it and say ‘I don’t wanna taste. I don’t wanna eat this.’ So it kind of be challenging just to, you know, get them to taste it.*”(Participant 1.3)

“*(kids won’t eat whole grain bread) ‘cause they’re used to the super soft super white super. Yeah. Fake bread.*”(Participant 5.2)

In part, providers blamed parents for failing to set a good example at home. They also talked about children bringing unhealthy foods with them to the child care. Overall, providers wanted more participation and support from parents.

“*Seeing what these kids come into the daycare center from home…what they bring in, hot Cheetos for breakfast. Instantly, you know, like to ‘Let me switch this out for a granola bar or a cereal bar’ just to give them something healthy and say ‘It’s OK to eat this every now and then,’ but watching this kid coming every day with McDonald’s, hot Cheetos, Hi-C juices, you know, like ‘OK, we have to do something about it.’*”(Participant 2.2)

### 3.4. Caring for Women and Families Who Choose to Breastfeed

Focus group participants were asked how they approached caring for infants of families that wanted to breastfeed or provide expressed milk. All providers discussed being supportive and open to parents who breastfeed or who provide expressed milk, and saw themselves as a resource to parents and families. Many providers talked about having private places for breastfeeding parents to nurse or pump. They reported encouraging parents to breastfeed and that they serve as a resource when families struggle or have questions about breastfeeding. Most providers were very comfortable in their knowledge about breastfeeding from personal experience.

“*My biggest that I have to share is, ‘this way hurts,’…and ‘I can’t hold him that way.’ Well, have you ever, you know, tried it this way? Have you ever used the football hold or the backwards way? And they’re like, ‘Oh, we can do it that way?’*”(Participant 5.2)

“*Usually if they ask or if they say something. Then I would, you know, give them some information.*”(Participant 5.7)

Participants talked about the importance of respecting parents’ wishes regarding breastfeeding. They saw their role as reassuring parents that their child is being cared for and fed accordingly.

“*It’s 100% the parents’ job to tell us what we need to be doing there…breastfeeding is one of the two things that I would listen to the parents and follow the parents’ wishes 100%.*”(Participant 5.4)

“*We’re there to just act as a second parent. You know, second caregiver, and we’re just going to follow their wishes whatever they want, so just support them.*”(Participant 5.6)

Providers felt that primary care doctors, the community, friends, and family should all provide support for families who choose to breastfeed. One provider said “I feel like it should be normalized. It’s 2020.” (Participant 5.6) A few providers talked about wishing that doctors would talk to parents more about the benefits of breastfeeding because providers did not want parents to think they were too intrusive in their decision-making surrounding breastfeeding.

Providers said they had access to plenty of resources such as lactation consultants, informational sheets, and personal knowledge and experience, even though none reported having formal breastfeeding training. However, providers also talked about wanting additional training. Specifically, they mentioned being interested in additional assistance in communicating with parents about breastfeeding and forming a breastfeeding plan and about cultural norms that may not align with best practices.

“*But you know that’s interesting because then you’re gonna have to do some training, because that’s a cultural thing (feeding cereal in a bottle). Now we’re talking about some cultures you know and so that’s deep rooted and so definitely we’d have to have research on that. You know, when we talking with parents about that? Why this is a best practice of not doing it, so yea, definitely we’d have to have some backup with that, yeah?*”(Participant 1.2)

### 3.5. Implementing and Supporting Screen-Time Best Practices

Participants were asked what helps them implement screen-time best practices, including what helps them limit screen time. Most providers felt they already follow best practices for limiting screen time and therefore did not need extra support. Some mentioned allowing screen time on a limited basis for exercise videos, yoga, or special occasions. In one focus group, providers talked about knowing that limiting screen time was important but they did not know specifically why. This uncertainty hampered their ability to communicate with parents about the importance of screen-time limits. Many were concerned about how much time older kids spend on their tablets for remote learning and in general. They talked about how some parents let their children stay on tablets or watch TV late into the night, which negatively impacts their sleep, and “in turn” their eating habits the next day.

“*That’s always an issue (sleepy children arriving in the morning). Keeping her up all night watching TV and so by the time they reach daycare, they’re too sleepy to even (eat) breakfast.*”(Participant 3.3)

“*You know what, they (parents) when they get home, that’s when they hand over their phone, like (other participant) was saying, they hand over the phone and the iPads to their children so they can get out of their way, so the parent can do what they gotta do. And when they come in, they be so tired and sleepy. I’m like, ‘so what time this child go to bed?’ (parent response) ‘I went to sleep I don’t know when I woke up at 4:00 o’clock they was still on.’*”(Participant 3.4)

Despite providers reporting that they follow best practices, several talked about having the TV on in the background during pick-up or drop-off or during mealtimes. They also mentioned feeling torn about children’s screen time since they also view it as educational.

“*He’s like obsessed with Sesame Street…I’m thinking like, well at least he’s learning. I can tell he’s not even two, and he knows so much for watching this stuff, and I feel like you know. I’m torn between should I allow him to it as much or take it from him?*”(Participant 1.4)

### 3.6. Needed Support

Participants were then shown a list of high-impact intervention best practices drawn from *Caring for Our Children: National Health and Safety Performance Standards; Guideline for Early Care and Education Programs, 4th Edition* [25]. When focus group participants were asked about the kind of support they needed and/or wanted in order to help them implement more of the best practices, providers expressed an interest in additional resources and sustained training and technical assistance. They wanted more funding to support nutrition efforts, and more recipe and snack ideas. They also wanted more participation and support from parents. 

When asked what resources they turn to when seeking information, they mentioned the CACFP program and monitoring staff, conventions and conferences, in-services, and free courses. They also mentioned the We Choose Health texting club, the internet, the Illinois Network of Child Care Resource and Referral Agencies (INCCRRA), and other providers themselves.

Focus group participants indicated that trusted sources for information on improving health practices were CACFP, Child Care Providers Network, INCCRRA, Gateways to Opportunity Illinois Professional Development System, Illinois Extension, and other outside organizations that send in behavioral specialists and nutritionists.

## 4. Discussion

In order to better understand barriers to implementing nutrition best practices in child care, we conducted focus groups with child care providers in Illinois. Our results indicated that providers experienced challenges including the high cost of nutritious food, children’s picky eating, and perceptions that parents did not set good examples at home. Many providers identified the Child and Adult Care Food Program (CACFP) as a critical resource in helping them implement best practices.

Improvements in the ECE food environment have been documented through strengthening state-wide policy and licensing requirements [43,44,45]. Somewhat surprisingly, our results indicate that participants did not think about licensing standards as a driving force behind their own practices and instead relied on experience. However, the guidelines and policies of CACFP were mentioned several times as being helpful in guiding practices. This is consistent with evidence that participation in CACFP has been associated with better nutrition best practices [46,47,48] such as more nutritious food and beverages being offered [49,50,51]. It has also been associated with healthier food consumption and may reduce the prevalence of overweight [42].

While center-based participation in CACFP has grown [52], participation by family child care providers has dropped in recent years [53]. There is some evidence that low-income children have less access to CACFP [54] and that rural providers have more challenges participating in the program [55]. Future research should address how to reduce the barriers to participation in CACFP particularly for providers of color, low-income, and/or located in rural areas.

Picky eating behaviors by children are a barrier to healthy diet consumption and are well documented as a challenge in ECE environments [56,57,58]. While picky eating is not well defined and there is no agreed upon operational definition [59], there is some emerging evidence that picky eating behaviors differ between home care and child care environments [60]. This was illustrated in a quote from a provider (above) who showed a parent a picture of their child eating a vegetable as proof that they would try new foods. Providers indicated that established nutrition best practices helped them get children to try and accept new foods.

Focus group participants felt that they were able to implement best practices for screen time, however some reported negative carryover from the impact of screen time in the home environment. The impact of the COVID-19 pandemic on increased screen time for all ages [61] is troubling given that screen time has been associated with obesity [62] and is a risk factor for severe obesity in children under 5 [63]. Children ages 3–7 across six countries reported increased screen time [64] and even children as young as 8–36 months are acquiring more screen time than before COVID-19 [65]. Future research should explore what providers know about why screen-time limits are recommended and to help develop training and tools to facilitate conversations between providers and parents about the importance of limiting screen time.

The negative impact of the COVID-19 pandemic on the child care system is well-documented [66,67]; however, the lasting impact on children’s food and nutrition environment is still emerging [12,68,69,70]. Many of the strides made in nutrition best practices are now at risk [67,71]. Providers frequently mentioned food costs as a barrier to nutrition best practices. This is troubling as higher food expenditures are associated with more nutritious and higher quality foods [72]. This, coupled with the fact that the United States Department of Agriculture (USDA) recently announced increases to the consumer price index (CPI), the highest in 42 years, may mean that accessing and paying for nutritious foods will be even more difficult for providers [73].

While providers indicated that they were supportive of parents who choose to breastfeed, none reported having formal breastfeeding training. Given that focus group participants indicated their knowledge and confidence in supporting breastfeeding came from experience, and that they were, on average, experienced, determining the training needs for new and less experienced providers is important. This underscores the need to provide training for providers on best practices including how to communicate with new parents who are transitioning back to the workforce.

Communicating and partnering with parents was something that providers indicated was sometimes difficult. In some instances, they felt that parents were barriers to supporting nutrition best practices through poor nutrition practices in the home. While the child care environment is important to supporting healthy eating behaviors, the home environment and the role of parents is also critical [22,74,75]. Developing ways for providers to effectively partner with parents is vital as nutrition/obesity prevention interventions that include parent engagement components have shown better outcomes [7,76,77]. Providers indicated wanting training on how to help them talk with parents about nutrition best practices together with why these practices were important for the health of their child.

Providers mentioned the need and desire for more training on multiple topics as has been previously discussed. Previous research has found that even when policies supporting nutrition best practices were successfully implemented, there was still a need for additional supports such as provider training [45]. It is encouraging that sources of nutrition information that providers trust are entities already in regular contact with child care providers, entities that administer licensing and programs such as CACFP, and who develop and deliver training (e.g., INCCRRA, CACFP). Organizations and others should work with child care providers when developing trainings to ensure they are effective and culturally relevant [78].

These focus group findings are currently being used by the ISPAN ECE working group to inform policy-change recommendations for licensing and program administration. IPHI is also using these findings to determine ways these barriers and challenges could be addressed by system-level supports and to inform strategic planning and to develop resources for technical assistance for providers to use to help child care providers make practice improvements. These efforts should be evaluated in order to further inform the field.

## 5. Limitations

These findings are limited in that they only represent child care providers in one state and the findings may not be generalizable to others in other geographical locations in the United States or to other countries. Illinois is one of six states in the United States that has led the adoption of high-impact obesity prevention standards into state-level licensing regulations [79]. Therefore, participants may be more familiar with, and exposed to, best practices through licensing standards. Providers may also have been more motivated to participate and discuss these topics due to their own interest in the focus group topics. Those interested in this topic and who attend a focus group may be more motivated to make practice improvements, so our findings may not generalize to all providers.

We did not specifically ask about differences in challenges implementing best practices across geography or child care setting and they did not emerge from the focus groups. However, given the findings of Dev et al. [80] that different settings pose differing challenges and barriers, future research should address this. While rural providers were represented, we were unable to include one focus group of providers from deeply rural areas due to the COVID-19 pandemic shift to online. Another limitation is that providers who participated in the focus groups were experienced and had worked as child care providers for a number of years. Future research should specifically target less experienced providers.

Physical activity is widely recognized as a contributor to childhood obesity and other poor health outcomes [81]. However, a limitation is that this paper focuses only on nutrition best practices in response to the focus of this special issue on the nutritional contributions to health outcomes in early childhood. We encourage future research to consider multiple contributors to health outcomes beyond nutrition such as physical activity [82].

## 6. Conclusions

In conclusion, the purpose of this study was to better understand the challenges and barriers that child care providers faced in implementing and integrating nutrition best practices in their child care. We found that providers struggle most with the high cost of nutritious food, children’s picky eating habits, and the perceived lack of aligned nutrition practices in the children’s own homes. This study also found that the COVID-19 pandemic had negative effects on children’s screen time that can have negative impacts on healthy nutrition behaviors. Providers indicated the high cost of nutritious food was a barrier which is troubling given the continuing increase in food costs. Child care providers indicated that they need and want more training and would like more support from parents. Organizations that support child care providers should ensure that training for providers addresses these needs. Providers view the CACFP as a vital resource in helping them understand and implement best practices, communicate with parents about supporting healthy eating habits, and reducing picky eating. These results underscore the importance of ensuring that all eligible children have access to participate in CACFP and that program reimbursement rates can adequately address the cost of providing nutritious food.

## Data Availability

The focus group guide and numeric data are available upon reasonable request to the authors. The focus group data are protected.

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
