# Peer review of "Informing State-Wide Coalition Efforts to Implement and Integrate Nutrition Best Practices in Early Care and Education: Focus Group Insights from Child Care Providers"

_ijerph, 2022, doi:10.3390/ijerph191610025_

Round 1

Reviewer 1 Report

The manuscript addresses a very important research problem whose resolution is worthy of generating new insights on the societal problem of obesity in children and adults.  The study has generated very important results whose application may contribute to the success of current nutrition outreach programs and better policy formulation informed by the results. While the manuscript is presented relatively well, there are few grammatical weaknesses that can be easily corrected. Apart from that observation, I have a few comments that can potentially improve the current manuscript draft. The comments are summarised as follows: 

Line 11:  The child care environment is critical to supporting healthy nutri-  tion as a majority of young children under 5 years of age are in out of home care

Comment: In the latter part of this sentence, it appears the words 'in out' are grammatically flawed. Thus, it is not clear to me what you mean? Please reformulate this sentence for maximising its meaning and the grammatical composition involved. 

Line 27-28: One of these diseases is obesity which remains at a stubbornly high level for children ages 2 to 5 [2–4]

Comment : The word 'ages' must be replaced with the word 'aged'  

Comment : Under the section that presents the results, very insightful comments are being made on the actual remarks that were made by members of the focus group discussions that took place. I am recommending that each remark or comments made are accompanied by for example, the number that represent the specific participant that made the comments. For example, under each narrative, you may choose to write the information source as follows:  

“He’s like obsessed with Sesame Street…I’m thinking like, well at least he’s learning. I can tell he’s not even two, and he knows so much for watching this stuff, and I feel like you know. I’m torn between should I allow him to it as much or take it from him?

                            Source: Participant No. AA1 or No. 9

Comment : Line 442-449: The sentences written in these lines represent the last paragraph in the section that discusses the results.  Upon reading this part, the sense I am getting is that the authors are presenting the study limitations. I am therefore recommending that this paragraph should be presented separately as Study Limitations. Please find an appropriate place in the manuscript to locate these limitations.  In the same section or subsection on Study Limitations, can you please elaborate on the generalisability of your results across geography and other similar contexts. Lastly,  the fact that the study collected primary data from only 25 respondents is also another limitation. Please include it. 

Comment: Line 430 -450: Under Conclusions

Your conclusion is too short. At the end of the introduction, you wrote as follows:  The specific goal of the focus groups was to better understand the needs, implementation practices, and barriers encountered by Illinois early childcare providers around nutrition, physical activity, and breastfeeding. 

Please add slightly more substance than is currently the case in your conclusion section that would better integrate better with the study purpose and the main findings and what can be synthesised out of these findingsSimilarly, please expand a little bit on the implications of the conclusions on the current nutrition programs.  Lastly, please make a distinction between recommendations for improved dietary lifestyles  amongst the vulnerable groups  and the recommendations that points out new areas for fruitful research in your chosen topic. 

The comments are given in a spirit of improving the manuscript. Best wishes. 

Author Response

Reviewer 1 -Thank you for your helpful comments and suggestions which have helped improve our manuscript. We have indicated how we addressed each of them in the attached file.

Reviewer 2 Report

Dear authors,

Firstly, congratulations on this original and worthy qualitative research topic performed. However, based on my point of view, I must regretfully inform you that your manuscript cannot be accepted for publication in the prestigious IJERPH at this moment.

In this sense, I would like to suggest some aspects to improve the significance of content, and quality of presentation of your results to increase the interest of the readers in the results of your study.

At the beginning of your introduction section, you show the relevance of “Poor diet is linked to diseases that are among the leading causes of death” and you focus on one of them, obesity, indicating that by 2030 one in two adults will be obese. As you know, obesity is a complex multifactorial disease, where physical activity plays a key factor related to chronic diseases. According to Booth and colleagues, physical inactivity is an actual contributing cause to at least 35 unhealthy conditions, including the majority of the 10 leading causes of death in the U.S. (Booth, F. W., Roberts, C. K., Thyfault, J. P., Ruegsegger, G. N., & Toedebusch, R. G. (2017). Role of Inactivity in Chronic Diseases: Evolutionary Insight and Pathophysiological Mechanisms %J Physiol Rev. 97(4), 1351-1402. doi:10.1152/physrev.00019.2016).

In this sense, you show in the fourth paragraph that “The Illinois State Physical Activity and Nutrition (ISPAN) program is an initiative funded by the Centers for Disease Control and Prevention to implement physical activity and nutrition interventions designed to tackle the root causes of obesity and make it easier for Illinoisans to live the healthiest lives possible” and then, in the fifth paragraph, you say: “ISPAN works through four strategies, only two of which are pertinent to this study and will be discussed here.”. However, at the end you say that your “specific goal of the focus groups was to better understand the needs, implementation practices, and barriers encountered by Illinois early childcare providers around nutrition, physical activity, and breastfeeding” Please, you must be coherent in your introduction´s statements according to the scientific literature and the method design/aims of your study and the third area of your focus group sessions ((3) implementing physical activity best practices (not ad-112 dressed in this paper) Why not?). In this sense, I highly recommend you introduce the crucial role of physical activity in your results because is very pertinent to your study and from my point of view it should be shown in your manuscript and not separate. In this way, I suggest discussing your results with the results shown in this study: González-Gross, M., Aparicio-Ugarriza, R., Calonge-Pascual, S., Gómez-Martínez, S., García-Carro, A., Zaragoza-Martí, A., Sanz-Valero, J., Wanden-Berghe, C., Martínez, J. A., Gil, Á., Marcos, A., Moreno, L. A., & On Behalf Of The Spanish Nutrition Society Señ (2021). Is Energy Expenditure or Physical Activity Considered When Energy Intake Is Measured? A Scoping Review 1975-2015. Nutrients, 13(9), 3262. https://doi.org/10.3390/nu13093262 (https://pubmed.ncbi.nlm.nih.gov/34579141/). On the contrary, I would consider that the main mistakes found before in this scientific study are been maintained in the studies to date.

Secondly, I think is mandatory to show in your results section a table summarising the results found in the primary and secondary coding analysis performed.

Finally, be careful with some grammar style aspects (e.g., n=23 (line 131); Black/White in appropriate inclusive language and in upper capital letters (line 133)).

I want to take this opportunity to thank you for your efforts and I consider the references cite and points offered by myself to accept your manuscript in this journal as a reviewer of the IJERPH journal.

Yours faithfully,

Author Response

Reviewer 2 - Thank you for your helpful suggestions and comments that have helped improve our manuscript. We have indicated how we addressed each of them in the attached file.

Round 2

Reviewer 2 Report

Dear authors,

I consider that the manuscript has improved, thank you for considering some of my suggestions about physical activity in your discussion and to specify as an imitation that you focus only on nutrition best practices in your paper. However, some commented aspects in the previous review must be corrected before accepting the manuscript to be published. Once again, I am concerned about the lack of scientific grammar style (e.g., n=23 (line 131)), please use “n” in lowercase. Also, please, use racial groups as nouns – Racial and ethnic terms should not be used in noun form (e.g., avoid “Hispanic or Latino” (line 132), “Black” and “White”) (line 133). Instead consider the adjectival form (e.g., Hispanic or Latino, White and Black individuals, children or participants…).

Furthermore, as you have mentioned in your response, please add as supplemental material the table with codes/themes and a description, adding the participant number in the illustrative quote, in the same way, that you did in the text of the manuscript (e.g. (Participant 1.2)).

From my point of view, with all these changes corrected, the manuscript is accepted to be published.

Yours faithfully,

Author Response

 Response to REVIEWER #2 Reviewer comments are in italics and our responses are indented.

I consider that the manuscript has improved, thank you for considering some of my suggestions about physical activity in your discussion and to specify as a limitation that you focus only on nutrition best practices in your paper. 

Thank you for your encouraging and positive feedback. We appreciate your suggestion about including the limitation of not addressing physical activity in our manuscript and believe it is stronger because of this addition.

However, some commented aspects in the previous review must be corrected before accepting the manuscript to be published. Once again, I am concerned about the lack of scientific grammar style (e.g., n=23 (line 131)), please use “n” in lowercase.

Thank you for pointing out that we had a capital N in line 131. This has now been corrected and changed to a lower case “n”.

Also, please, use racial groups as nouns – Racial and ethnic terms should not be used in noun form (e.g., avoid “Hispanic or Latino” (line 132), “Black” and “White”) (line 133). Instead consider the adjectival form (e.g., Hispanic or Latino, White and Black individuals, children or participants…).

Thank you for making this suggestion. The manuscript now reads “Thirty six percent of participants self-identified as Hispanic or Latino individuals (n=8), 50% of participants identified as Black individuals (n=11), 36% of participants identified as White individuals (n=8), 4.5% of participants identified as individuals of multiple races (n=1) and 9% of participants selected “other” (n=2).”

Furthermore, as you have mentioned in your response, please add as supplemental material the table with codes/themes and a description, adding the participant number in the illustrative quote, in the same way, that you did in the text of the manuscript (e.g. (Participant 1.2)).

Thank you for suggesting that we include the table of codes/themes (with participant identifiers added) in the supplemental material. We have provided this believe this will be helpful to the readers. We have referenced this at the end of the Methods section [line 128-129].